# Different Oxidation Pathways of 2-Selenouracil and 2-Thiouracil, Natural Components of Transfer RNA

**DOI:** 10.3390/ijms21175956

**Published:** 2020-08-19

**Authors:** Katarzyna Kulik, Klaudia Sadowska, Ewelina Wielgus, Barbara Pacholczyk-Sienicka, Elzbieta Sochacka, Barbara Nawrot

**Affiliations:** 1Centre of Molecular and Macromolecular Studies, Polish Academy of Sciences, Sienkiewicza 112, 90-363 Lodz, Poland; kpieta@cbmm.lodz.pl (K.K.); ms@cbmm.lodz.pl (E.W.); 2Institute of Organic Chemistry, Lodz University of Technology, Zeromskiego 116, 90-924 Lodz, Poland; klaudia.sadowska@dokt.p.lodz.pl (K.S.); barbara.pacholczyk@p.lodz.pl (B.P.-S.); elzbieta.sochacka@p.lodz.pl (E.S.)

**Keywords:** 2-thiouridine, 2-selenouridine, 2-thiouracil, 2-selenouracil, oxidative stress, modified nucleoside, tRNA

## Abstract

Sulfur- and selenium-modified uridines present in the wobble position of transfer RNAs (tRNAs) play an important role in the precise reading of genetic information and tuning of protein biosynthesis in all three domains of life. Both sulfur and selenium chalcogens functionally operate as key elements of biological molecules involved in the protection of cells against oxidative damage. In this work, 2-thiouracil (S2Ura) and 2-selenouracil (Se2Ura) were treated with hydrogen peroxide at 1:0.5, 1:1, and 1:10 molar ratios and at selected pH values ranging from 5 to 8. It was found that Se2Ura was more prone to oxidation than its sulfur analog, and if reacted with H_2_O_2_ at a 1:1 or lower molar ratio, it predominantly produced diselenide Ura-Se-Se-Ura, which spontaneously transformed to a previously unknown Se-containing two-ring compound. Its deselenation furnished the major reaction product, a structure not related to any known biological species. Under the same conditions, only a small amount of S2Ura was oxidized to form Ura-SO_2_H and uracil (Ura). In contrast, 10-fold excess hydrogen peroxide converted Se2Ura and S2Ura into corresponding Ura-SeO_n_H and Ura-SO_n_H intermediates, which decomposed with the release of selenium and sulfur oxide(s) to yield Ura as either a predominant or exclusive product, respectively. Our results confirmed significantly different oxidation pathways of 2-selenouracil and 2-thiouracil.

## 1. Introduction

Sulfur and selenium elements are natural components of living organisms in all three domains of life. Among them, the most commonly known are cysteine (Cys) and selenocysteine (Sec) building blocks of proteins and 2-thiouridine (S2U) and 2-selenouridine (Se2U) present in transfer RNAs (tRNAs). Both elements functionally operate as key elements of the enzymes involved in the protection of cells against oxidative damage. These elements are active redox components involved in thiol/disulfide exchange, reactive oxygen species (ROS) metabolism, and redox homeostasis [1,2,3,4,5,6,7,8]. Although selenium and sulfur have similar chemical properties [9], they differ significantly with respect to their polarizability and redox properties [10]. For example, due to weak Se–O π-bonding, Se-oxides are more easily reduced than S-oxides. This so-called “selenium paradox” [11] and other experimental data have allowed the suggestion that selenium-bearing compounds are superior ROS scavengers due to their unique ability to react with oxidizing species in a reversible manner, in contrast to their sulfur analogs [12].

Unfortunately, limited data are available in the literature on the susceptibility of S2U/Se2U to oxidation. In the course of our research on the functional properties of chalcogen-containing nucleotides present in transfer RNAs, we have shown that in the presence of hydrogen peroxide at high concentrations or in the presence of oxone^®^, 2-thiouridine (S2U) either as a nucleoside or in an RNA chain undergoes efficient oxidative desulfuration, furnishing uridine (U) and/or 4-pyrimidinone ribonucleoside (H2U) [13,14]. The ratio of products depends on buffer pH, the concentration of reagents, and the electronic properties of a substituent at the C5 position of the nucleobase [15,16]. The reaction proceeds through sulfur-oxidized intermediates (bearing a sulfenic, sulfinic, or sulfonic moiety) and is catalyzed in vitro by cytochrome C [17]. The S2U-RNA → H2U-RNA transformation leads to the loss of U-A base pairing in RNA duplexes [14]. Although this kind of tRNA damage has not been confirmed in cells and is still a subject of research, one can consider that it may impair tRNA function during protein synthesis due to the disruption of codon-anticodon interactions.

Recently, Hondal’s group investigated the oxidative transformation of 2-thio-and 2-selenouracils substituted at position 5 with a nonnative substituent (introduced to enhance nucleobase solubility in aqueous media), which is a carboxyl group directly attached to C5 (c5S2Ura and c5Se2Ura), and demonstrated that 5-carboxy-2-thiouracil in the presence of a 1:1 molar ratio of H_2_O_2_ underwent irreversible desulfuration, leading to 5-carboxyuracil. In contrast, the 2-seleno-analog, under the same conditions, was converted into the corresponding diselenide and seleninic acid, which, in a reducing environment, could be converted back to the starting 5-carboxy-2-selenouracil [18]. This result allowed them to conclude that selenium-containing biomolecules are resistant to permanent oxidation, and this is the main reason for naturally occurring selenium in both 2-selenouridine- and Sec-containing proteins. This conclusion complementary to the latest data on the role of seleno modifications in tRNA ensuring the fidelity of the reading of synonymous 3′-G ending codons [19], prompted us to perform more detailed studies on the oxidation of 2-thiouracil (S2Ura) and 2-selenouracil (Se2Ura), especially in light of the recently discovered “oxidative desulfuration” of S2U, leading predominantly to 4-pyrimidinone derivatives [13,14,15,16,17].

In work presented here, we treated 2-selenouracil (Se2Ura, **1a**) and 2-thiouracil (S2Ura, **1b**) with hydrogen peroxide under different conditions (at different concentrations and various pH values) and identified intermediates and final products at several time points using ^1^H NMR spectroscopy and ultra-performance liquid chromatography coupled with high-resolution mass spectrometry and photodiode array detection (UPLC-PDA-ESI(−)-HRMS). The obtained data allowed us to propose possible transformation paths for selenium- and sulfur-containing uracils and to characterize their redox properties.

## 2. Results

### 2.1. General Approach for Analysis of the Course of Oxidation of 2-Selenouracil (***1a***) and 2-Thiouracil (***1b***) and Identification of the Reaction Products

To obtain data on a sequence of events during oxidation of 2-selenouracil (**1a**, Se2Ura) and 2-thiouracil (**1b**, S2Ura), 10 mM solutions in phosphate buffer at pH 5.0, 7.4, or 8.0 or in water were reacted at room temperature (r.t.) with 5, 10, or 100 mM hydrogen peroxide (1:0.5, 1:1, and 1:10 molar ratio, respectively). The reaction progress (from 1 min to 24 h) and structural data on the intermediates/products were gathered from ^1^H NMR and UPLC-PDA-ESI(−)-HRMS measurements. The relative content of the detected compounds was calculated using integrations of ^1^H NMR signals for non-exchangeable H5 and H6 protons (δ range 8.8–5.2 ppm). The UPLC-ESI(−)-MS retention times (Rt, min), *m/z* values of the ions, which correspond to the deprotonated molecules, λ_max_ in UV/VIS spectra (nm), and ^1^H NMR chemical shifts (δ, ppm) for H5 and H6 protons and coupling constants *J_H6-H5_* for all identified compounds are presented in Table 1. Spectral data for all identified compounds are given in the Appendix A.

### 2.2. Analysis of the Oxidation Course of Se2Ura (***1a***)

#### 2.2.1. The Reaction of Se2Ura and H_2_O_2_ at a 1:1 Molar Ratio, pH 7.4

Preliminary experiments showed that to obtain reliable data on possibly all intermediates, the oxidation process should be performed at a pH close to neutral, so pH 7.4 was selected. A set of ^1^H NMR spectra (Figure 1) was acquired in the 24-h time course and compared with the spectrum of substrate **1a**. At similar time points, the UPLC-PDA-ESI(−)-HRMS data (Figure 2a) were collected. Dynamic changes were observed in the first stage of the reaction course (t < 60 min, see Figure 2b), since over the first 1–2 min, the resonances and MS signals characteristic of **1a** (δ 7.61 ppm and 6.21 ppm; *m/z* 174.9415) disappeared almost completely, while a new compound appeared, which was identified as diselenide **2a** (Ura-Se-Se-Ura) (δ 7.88 ppm and 6.26 ppm; *m/z* 348.8748) (Scheme 1). Later (from 5 to 30 min), the diselenide content gradually decreased, and after ca. 2 h, **2a** disappeared almost completely (see Figure 2a,b). Simultaneously, a new, stable product gradually appeared, for which two pairs of resonance signals were observed, i.e., δ 8.09 and 7.85 ppm for two different H6 protons and δ 6.42 and 5.98 ppm for two H5 protons. The *m/z* 205.0362 value in the mass spectrum for this product was lower than that for diselenide **2a** (*m/z* 348.8748) but higher than that for **1a** (*m/z* 174.9415). Detailed analysis (UPLC-ESI(−)-MS/MS fragmentation, Figure 3c) revealed that a selenium-free, two-ring compound **8** (Scheme 1) was formed, which, after 24 h, constituted 87% of the reaction products. The traces of compound **7a** (*m/z* 268.9584, Figure 3a), which might be a selenium-bearing precursor of **8**, were observed in the UPLC-HRMS measurement (Δ*m/z* 63.9222 between the ions related to the deprotonated molecules of **7a** and **8** conformed to the Se → O replacement) but not in the ^1^H NMR spectrum. The N1_a_-C2_b_ covalent bonding between two rings was confirmed by the presence of fragmentation ions at *m/z* 163.0305 and 134.0355 for **8** and at *m/z* 225.9521 and 162.0300 for **7a** in the collision-induced dissociation (CID) spectra. The pH-dependent UV measurements of **8** in the pH range of 3–10 are shown in Figure 2c.

The structure of **8** was confirmed by NMR analysis in the 2D HMBC (Heteronuclear Multiple Bond Coherence) spectrum, where the correlation between the H6_a_ proton (δ 7.85 ppm) and the C2_b_ carbon (δ 156.10 ppm) indicated a covalent bond between N1_a_ and C2_b_. Moreover, the same carbon atom C2_b_ also had a three-bond correlation with proton H6_b_ at 8.10 ppm (see Figure 4a). The ^1^H and ^13^C NMR spectra recorded for **8** are shown in Appendix A.

Another minor product (ca. 5%), identified in the reaction mixture after 24 h, was found to be three-ring derivative **9** (see Scheme 1). Its structure was confirmed by UPLC-HRMS analysis (*m/z* 299.0534) and ^1^H NMR resonances for three different H5 atoms and three different H6 atoms (Figure 1 and Figure 2a, Table 1). The UV spectra over a broad pH range (see Appendix A) were almost identical, and only one band with λ_max_ 257 nm at pH 3 shifted to 245 nm at pH 10. The collision-induced dissociation (CID) spectrum of **9** (Figure 3d) confirmed the presence of the three six-membered rings a, b, and c connected by N1_a_-C2_b_ and N1_b_-C2_c_ covalent bonds. The structure was also confirmed by the 2D HMBC spectrum shown in Figure 4b, where three bond correlations between C2_a_ and H6_a_ (δ 7.90 ppm) and H6_b_ (δ 8.46 ppm) were found. Similarly, correlations between C2_c_ (δ 156.79 ppm) and H6_b_ (δ 8.46 ppm) and H6_c_ (δ 8.02 ppm) were identified.

Our approach also allowed the identification of triselenide **3a** (Ura-Se-Se-Se-Ura, *m/z* 428.7909) with its highest abundance (ca. 5%) after 10 min and final disappearance after 2 h. Very weak signals for **1a** were noted after ca. 60 min (Figure 2a), indicating its partial restoration; however, its content after 24 h was virtually negligible. At an early time point (5 min), small amounts of seleninic acid **4a** (*n* = 2, Ura-SeO_2_H, *m/z* 206.9307) appeared, for which resonances at δ 8.09 ppm and δ 6.48 ppm were noted. A minute amount of very reactive selenenic acid, Ura-SeOH (**4a**, *n* = 1, *m/z* 190.9367), was observed at a rather late stage of ca. 60 min, so its appearance might be attributed to the disproportionation of an intermediate compound(s). After 24 h, 8% uracil (**5**) was identified, and the residual amount of inorganic H_2_SeO_3_ was observed in UPLC-HRMS analysis.

In an analogous reaction carried out in phosphate buffer at pH 8.0, the stability of diselenide **2a** decreased remarkably since after 1 min, the efficient formation of **8** was observed (see NMR analysis Appendix A). Compared to that detected for the reaction at pH 7.4, more seleninic acid **4a** (*n* = 2) was detected in the reaction mixture after 24 h. The formation of the three-ring product **9** was also observed. After 24 h, the reaction mixture consisted of compounds **8** (48%), **9** (17%), **5** (6%), **4a** (*n* = 2) (15%), and reconstituted **1a** (14%) (Scheme 1).

Finally, the reactions were performed under slightly acidic conditions (in phosphate buffer at pH 5.0 and in deionized water at pH 6.5). The relevant ^1^H NMR spectra (Appendix A) indicated that after 2 min at pH 5.0, diselenide **2a** was not detected, and the two-ring products **8** (46%) and uracil (**5**) (54%) were highly abundant. Interestingly, neither three-ring compound **9** nor restored **1a** were found. For the reaction carried out in the water, the reaction course was quite similar, and **8** and **5** were obtained in 37% and 63% yield, respectively (Appendix A). This change in the product ratio was not unexpected since the reacting mixture became more acidic (to pH 1.0–2.5) as inorganic H_2_SeO_3_ was released.

#### 2.2.2. The Reaction of Se2Ura and H_2_O_2_ at a 1:0.5 Molar Ratio and pH 7.4

The reaction of **1a** (10 mM) was carried out with 0.5 equivalents (5 mM) of H_2_O_2_ in phosphate buffer at pH 7.4 and monitored by ^1^H NMR (see Figure 5 and Appendix A). Interestingly, in this case **1a** was almost immediately converted to diselenide **2a** as a sole product (see Figure 5), and only a small amount of **4a** (*n* = 2, Ura-SeO_2_H) was observed after longer reaction times. Over the next 60 min, the formation of the two-ring and three-ring derivatives **8** and **9** was observed. After 24 h, the integration of ^1^H NMR signals indicated the formation of **8** (57%), **9** (26%), and **5** (6%), as well as restored **1a** (11%) (Scheme 2).

#### 2.2.3. The Reaction of Se2Ura and H_2_O_2_ at a 1:10 Molar Ratio and pH 7.4

A 10-fold molar excess of H_2_O_2_ towards Se2Ura (**1a**) caused almost immediate disappearance of **1a** (Appendix A) and formation of the seleninic acid derivative **4a** (*n* = 2, Ura-SeO_2_H) and diselenide **2a** in yields of ca. 16% and 75%, respectively. The ^1^H NMR spectrum recorded after 5 min indicated an increased amount of **4a** (*n* = 2) at the expense of diselenide **2a**. After 24 h, the mixture consisted of uracil (**5**, 89%, from the decomposition of Ura-SeO_2_H) and the two-ring derivative **8** (11%, from the rearrangement of diselenide **2a**) (Scheme 3). Interestingly, no three-ring product was detected. Only traces of **7a,** as well as inorganic H_2_SeO_3_ and H_2_SeO_4_, were detected by UPLC-HRMS (Appendix A).

### 2.3. Analysis of the Oxidation Course of S2Ura (***1b***)

#### 2.3.1. The Reaction of S2Ura and H_2_O_2_ at a 1:1 Molar Ratio and pH 7.4.

The reaction of S2Ura (**1b**, 10 mM) with hydrogen peroxide at a 1:1 molar ratio (Scheme 4, Figure 6 and Figure 7a) was much slower than that of the selenium analog **1a**, and after 24 h, ca. 45% of the substrate remained unchanged. Formation of sulfinic acid **4b** (*n* = 2, Ura-SO_2_H) (δ 8.07 and 6.49 ppm, *m/z* 158.9865) was noted immediately after mixing the reactants. Its maximum concentration occurred after approximately 2 h and decreased to approximately 34% after 24 h. In contrast to the oxidation of **1a**, where at the early time points, diselenide **2a** was the main intermediate, disulfide **2b** (δ 7.88 and 6.22 ppm, *m/z* 252.9853) was present in a small amount only (0.4%), and it was stable until the end of the experiment. At consecutive time points, other low-abundance compounds were identified: sulfonic acid **4b** (*n* = 3, Ura-SO_3_H, δ 8.02 and 6.44 ppm, *m/z* 174.9820, approx. 5%), sulfenic acid **4b** (*n* = 1, Ura-SOH, *m/z* 142.9921), 4-pyrimidinone (**6**, δ 8.02 and 6.55 ppm, *m/z* 95.0239), trisulfide **3b** (*m/z* 284.9576), and the 2-thiouracil member of the Bunte salt family **10** (δ 7.82 and 5.96, *m/z* 206.9539, see Appendix A), as well as the two-ring derivatives **7b** (*m/z* 221.0135, see Figure 3b) and **8** (*m/z* 205.0362). After 24 h, the reaction mixture contained three major products (Scheme 4): substrate **1b** (37%), sulfinic acid **4b** (*n* = 2, 34%), and uracil (**5**, 17%). The spectroscopic and chromatographic data for the identified compounds are given in Table 1, and ^1^H NMR spectra, as well as the time course of the oxidation reaction (Appendix A), are included in the Appendix A.

#### 2.3.2. The Reaction of S2Ura and H_2_O_2_ at a 1:10 Molar Ratio and pH 7.4

The reaction of **1b** with a 10-fold molar excess of hydrogen peroxide was relatively fast, and the substrate disappeared after 2 h (Scheme 5, Appendix A and Figure 7b). Initially, mainly sulfinic acid intermediate **4b** (*n* = 2; Ura-SO_2_H) (*m/z* 158.9865) was observed, but after 30 min, its content decreased, while the content of uracil increased. A minute amount of disulfide **2b** (*m/z* 252.9853) was detected by ^1^H NMR. The content of sulfonic acid **4b** (*n* = 3; *m/z* 174.9820) increased over the first hour of the reaction course. Traces of sulfenic acid **4b** (*n* = 1; *m/z* 142.9921), 4-pyrimidinone (**6**, *m/z* 95.0239), and Bunte salt **10** (*m/z* 206.9539), as well as two-ring compound **7b** (*m/z* 221.0135), were identified in UPLC-HRMS analysis. After 24 h, only the resonance signals of uracil **5** were present in the NMR spectrum, but traces of sulfinic and sulfonic acids were still detected in the UPLC-MS chromatogram (Figure 7b).

## 3. Discussion

2-Selenouracil (**1a**, Se2Ura) and 2-thiouracil (**1b**, S2Ura) are relatively rare nucleobases present in transfer RNAs (http://modomics.genesilico.pl [20], http://mods.rna.albany.edu [21]). Within presumably acceptable simplification (no specific substituent at position C5), these compounds may be used as models in preliminary investigations of the processes occurring with tRNA under oxidative stress conditions. Both compounds are sufficiently soluble in aqueous solutions for ^1^H NMR spectroscopy (in D_2_O) and advanced high-resolution LC/MS analysis.

Generally, our investigations have shown that 2-selenouracil (**1a**) and 2-thiouracil (**1b**) have significantly different redox properties (Scheme 6, the red path is preferable for selenium components, and the blue path is preferable for sulfur components). Compound **1a** is extremely prone to H_2_O_2_-assisted oxidation, plausibly resulting in the formation of selenenic acid **4a** (*n* = 1, Ura-SeOH). This very reactive compound is not identified in the first few minutes of the reaction (it is detected later on in a relatively complex reaction mixture) since, probably, it reacts rapidly with **1a**, producing diselenide **2a**. This path is in agreement with the results of the reaction carried out with 0.5 equivalents of H_2_O_2_, where **2a** is almost exclusively formed (Figure 5 and Appendix A) from **1a** and **4a** (*n* = 1). According to our assumption, this pathway of oxidation includes the two following steps: (i) Ura-SeH → Ura-SeOH and (ii) Ura-SeH + Ura-SeOH → Ura-Se-Se-Ura.

On the other hand, according to the literature data [22], diselenide R-Se-Se-R under alkaline conditions may hydrolyze to R-SeH and selenenic acid R-SeOH, and the latter, as an extremely unstable derivative, may disproportionate to produce the R-SeH substrate and seleninic acid R-SeO_2_H (see Scheme 7). This mechanism may explain the partial reconstitution of Ura-SeH (**1a**) from diselenide **2a** (without a reducing environment).

We also demonstrate that the route for reconstitution of **1a** from **2a** is limited due to the consumption/depletion of diselenide **2a** by its rapid intramolecular rearrangement, leading to the selenium-containing intermediate **7a** called “Jaffe’s base” (Scheme 1) [23]. In aqueous environments, this compound easily loses the selenium atom to form stable compound **8**. The relevant mechanism is similar to that proposed for the formation of “Jaffe’s base” during the oxidation of ethylene thiourea (Scheme 8a) [24]. This reaction, obviously, does not take place on the level of tRNA, when Se2Ura is built in the nucleoside moiety. However, one cannot exclude the possibility that the Se2U-tRNA is metabolized (nucleolytically degraded [25]) to Se2Ura, which, in oxidative stress, might be transformed into two- and three-ring products described herein.

There are some reports suggesting that 2-selenouracil derivatives substituted at the C6 position with alkyl groups react with iodine to form corresponding R6Ura-SeI_2_ complexes (where R = methyl, ethyl, *n*-propyl, or *iso*-propyl), which may further react with the starting selenouracil R6Se2Ura to form the two-ring products R6Se2Ura-R6Ura, which are deprived of the selenium atom in ring b [26]. As shown by crystallographic analysis, these products have a covalent bond between N3_a_ of one R6Se2Ura (a) ring and the C2_b_ atom of the second R6Se2Ura molecule (b). Ultimately, these compounds spontaneously transform into four-ring derivatives in which the two-ring components are linked with a diselenide bridge. Alternatively, in aqueous solutions, these compounds undergo deselenation and cleavage of the N3_a_-C2_b_ bond, leading to the uracil molecule being substituted with an alkyl group at position 6.

In contrast to the cited two-ring compound R6Se2Ura(N3_a_)-(C2_b_)R6Ura, the two-ring compound **8** reported here has a spanning bond between the C2_b_ and N1_a_ atoms, as documented by 2D NMR studies (Figure 4a). If the amount of oxidant is stoichiometrically limiting (see data for a 1:0.5 molar ratio of reactants at pH 7.4), or the reaction occurs under more basic conditions (pH 8), derivative **8** could be transformed to three-ring product **9** with a yield up to 26% of the total content. At first, we have supposed that compound **9** is a product of the condensation of **8** with **1a**, with the departure of hydrogen selenide as a second product (according to the mechanism proposed in Appendix A). If so, this reaction is expected to run between **8** and **1a** without any additional reagent. However, as shown by the ^1^H NMR analysis of **8** in the mixture with an excess of **1a** (0.4:1 molar ratio), only signals of traces of compound **9** are noted after 24 h. However, the prolonged incubation time of this mixture up to 7 days allowed to increase the content of **9** up to ca 14%, while the content of **1a** clearly decreased to 32% (Appendix A). Interestingly, when the same reaction was tested (**8** and **1a**, 0.4:1 molar ratio) but with the addition of 0.5 eq. of hydrogen peroxide over **1a**, the formation of **9** after 1 h was observed, to reach finally the yield of ca. 14% after 24 h (see Appendix A). Thus, we conclude that compound **9** is mostly a product of the condensation of **8** and **2a**, as shown in Scheme 8b, in which 2-selenouracil **1a** and selenium are released, Thus, the reconstituted 2-selenouracil, present as a final product in the reaction, shown in Scheme 2, may originate from the diselenide reacting with compound **8**.

In general, we can summarize that 2-selenouracil (**1a**) in the presence of a stoichiometrically limiting amount of the oxidant is preferably transformed to diselenide **2a**, which spontaneously rearranges to the two-ring product **8**, which, in turn, reacts with the remaining diselenide **2a** to yield **9** (red path, Scheme 6).

This type of rearrangement has not been reported earlier for the oxidation experiments carried out with c5Se2Ura [18]. In this cited work, the oxidation of c5Se2Ura furnishes oxidized forms of c5Se2Ura and uracil. In our case, in contrast, the main component is the two-ring product **8**. This inconsistency may originate from the use of Se2Ura bearing a carboxyl substituent in position C5, which exerts an electron-withdrawing effect (both by induction and resonance). This effect is evidenced by the change in the p*K*a value from 7.18 for Se2Ura (**1a**) to 7.11 for c5Se2Ura. A similar effect has been observed for 2-thiouracils, where the p*K*a value changes from 7.75 for S2Ura (**1b**) to 7.68 for c5S2Ura [27].

The route leading to diselenide **2a** and its rearrangement to **7a**, **8,** and **9** is less favorable if the oxidant is present in high excess (see Appendix A). In these conditions, the prevalent formation of seleninic acid **4a** (*n* = 2, Ura-SeO_2_H) is noted, which, in aqueous solutions, undergoes water-assisted hydrolysis, resulting in the formation of uracil (**5**) (89%) accompanied by elimination of selenium (IV) oxide, which reacts with water to form H_2_SeO_3_ (blue path, Scheme 6). Besides, it is worth mentioning that, according to published data, in the presence of excess hydrogen peroxide, the diselenides are further oxidized to selenoseleninate derivatives (-Se(O)-Se-), which are unstable under basic conditions [28,29,30]. These compounds are hydrolyzed to seleninic acid (RSeO_2_H) and selenol (RSeH), while, in acidic environments, selenenic acid (RSeOH) is formed [31]. Our efforts to confirm such transformation pathways have been unsuccessful, probably due to the low stability of the selenoseleninate derivative.

It is noteworthy that, in the Hondal team’s work [18], treatment of 5-carboxy-2-selenouracil (c5Se2Ura, 100 mM) with 1 molar equivalent of H_2_O_2_ after 18 h led to a product resonating at δ 1273 ppm in ^77^Se NMR, which was assigned as seleninic acid c5Ura-SeO_2_H. In our reaction carried out in phosphate buffer at pH 7.4, seleninic acid Ura-SeO_2_H (**4a**, *n* = 2) is present only in the first several minutes, while inorganic seleninic acid H_2_SeO_3_ is present after 1 h (see Figure 2a). In addition, similar chemical shifts are known to be associated with Na_2_SeO_3_ (1263 ppm) and H_2_SeO_3_ (1300 ppm) [32].

Unlike 2-selenouracil (**1a**), 2-thiouracil (**1b**) is less prone to oxidation with H_2_O_2_. The reaction is remarkably slower, and a 10-fold molar excess of oxidant leads to predominant conversion to uracil (**5**) (Scheme 5). With equal or stoichiometrically limiting amounts of H_2_O_2_, ca. 40% of **1b** remains intact. The identified intermediates include sulfenic acid **4b** (*n* = 1, Ura-SOH), sulfinic acid **4b** (*n* = 2, Ura-SO_2_H), and sulfonic acid **4b** (*n* = 3, Ura-SO_3_H) (Scheme 6, blue). Notably, among those three acid forms, sulfinic acid (Ura-SO_2_H) is the most abundant, as shown by ^1^H NMR analysis (having the highest signal integration and the longest time, the signals remained in the reaction mixture). Similar to sulfonic acid (Ura-SO_3_H), this compound can eliminate sulfur oxide upon nucleophilic substitution of a water molecule at the C2 position of the nucleobase ring. Although the small amount of 4-pyrimidinone (**6**) is identified, this process is marginal in comparison to the previously reported data for 2-thiouridines [13,14,15,16,17]. Traces of disulfide **1b**, trisulfide **1c**, or bicyclic compounds **7b** and **8** (detected by UPLC-HRMS) indicate that while the responses of S2Ura and Se2Ura to hydrogen peroxide are common, their different redox properties are decisive for the preferred paths and the final content of oxidized products.

## 4. Materials and Methods

### 4.1. Methods and Instrumentation

#### 4.1.1. NMR Spectroscopy

^1^H, ^13^C, COSY (COrrelation SpectroscopY), HMQC (Heteronuclear Multiple Quantum Correlation) and HMBC NMR spectra were recorded on a Bruker Avance 700 MHz spectrometer. The NMR spectra for ^1^H and ^13^C were recorded at 700 MHz and 176 MHz, respectively. Chemical shifts (δ) are reported in ppm, and the signal multiplicities are described as s (singlet), d (doublet), t (triplet), q (quartet), m (multiplet). Coupling constants are reported in hertz (Hz).

#### 4.1.2. Ultra-Performance Liquid Chromatography Coupled with a High-Resolution Mass Spectrometry and Photodiode Array Detection (UPLC-PDA-ESI(−)-HRMS)

The identification of the reaction products was carried out using an ACQUITY UPLC I-Class chromatography system equipped with a photodiode array detector with a binary solvent manager (Waters Corp., Milford, MA, USA ) coupled with an SYNAPT G2-S*i* mass spectrometer equipped with an electrospray source and quadrupole-Time-of-Flight mass analyzer (Waters Corp., Milford, MA, USA). An Acquity HSS T3 1.8 μm column (100 × 2.1 mm) (Waters Corp., Milford, MA) thermostated at 30 °C was used for the chromatographic separation of the analyte. A gradient program was employed with the mobile phase combining solvent A (10 mM CH_3_COONH_4_) and solvent B (50% CH_3_CN in 10 mM CH_3_COONH_4_) as follows: 10% B (0–1.0 min), 10–95% B (1.0–3.5 min), 95–99% B (3.5–4.0 min), 95–10% B (4.0–4.1 min), and 10–10% B (4.1–6 min). The flow rate was 0.2 mL/min, and the injection volume was 1 μL.

For mass spectrometric detection, the electrospray source was operated in a negative, high-resolution mode at a 50,000 FWHM resolving power of the TOF analyzer. To ensure accurate mass measurements, data were collected in centroid mode, and mass was corrected during acquisition using leucine encephalin solution as an external reference, Lock-Spray^TM^, (Waters Corp., Milford, MA, USA), which generated reference ion at *m/z* 554.2615 ([M-H]) in negative ESI mode. The optimized source parameters were: capillary voltage 3 kV, cone voltage 20 V, source temperature 90 °C, desolvation gas (nitrogen) flow rate 600 L/h with the temperature 350 °C, nebulizer gas pressure 6.5 bar. Mass spectrometer conditions were optimized by direct infusion of the standard solution. Mass spectra would be recorded over an *m/z* range of 100 to 1200. Collision-induced fragmentation experiments were performed using argon as the collision gas. The collision energy was ramped from 15 to 35 eV.

The PDA spectra were measured over the wavelength range of 210–400 nm in steps of 1.2 nm.

The results of the measurements were processed using the MassLynx 4.1 software (Waters) incorporated with the instrument.

#### 4.1.3. Ultraviolet Spectroscopy Measurements (UV)

UV spectra were recorded on a Specord^®^ 50 plus spectrophotometer. Samples were prepared by dilution of 4 μL of stock compounds solution (stock solution-ca 1 mg of compound in 1 mL water) in 996 μL of buffer solutions (1 mM HCl at pH 3.0, 67 mM phosphate buffer at pH 5.0, 6.0, 6.5, 7.0, 7.5, 8.0, 0.1 mM NaOH at pH 10).

### 4.2. Experimental Section

#### 4.2.1. Materials

All materials, including 2-thiouracil, were purchased from Sigma Aldrich (St. Louis, MO, USA) or TCI Europe *n*.V., (Zwijndrecht, Belgium).

#### 4.2.2. Synthesis of 2-Selenouracil

The synthesis of 2-selenouracil was done according to the published procedure [33], with slight improvement, and is described in Appendix A.

#### 4.2.3. ^1^H-NMR Analysis of Oxidation Assays of **1a** and **1b**

A 10 mM solution of either **1a** or **1b** was prepared in 67 mM phosphate buffer (pH 7.4, pH 8.0, pH 5.0) or deionized water (using D_2_O). The first ^1^H NMR spectrum was acquired to establish the initial point. Compounds **1a** or **1b** were treated with 1 or 10 equivalents of H_2_O_2_. The reactions were monitored by ^1^H NMR, and the spectra were acquired after 1 or 2 min, 5 min, 20 min, 30 min, 60 min, 120 min, and 24 h.

#### 4.2.4. UPLC-PDA-ESI (−)-HRMS Analysis of the Oxidation Assays of **1a** and **1b**

The 10 mM solutions of either **1a** or **1b** were prepared in 67 mM phosphate buffer (pH 7.4) and then were treated with 1 or 10 equivalents of H_2_O_2_. The reactions were monitored by UPLC-PDA-ESI (−)-HRMS, and the spectra were acquired after 1, 5, 10, 30, 60, and 120 min and 24 h.

## 5. Conclusions

By a series of oxidation experiments carried out on 2-seleno- and 2-thiouracil (**1a** and **1b**), we demonstrated that Se2Ura is more prone to oxidation than the sulfur analog. In the first step of oxidation of **1a** with H_2_O_2_ (1:0.5 molar ratio), diselenide **2a** is exclusively formed, which is, as we suggest, the product of the condensation of selenenic acid (Ura-SeOH) and Ura-SeH. Diselenide **2a** spontaneously undergoes Jaffe’s rearrangement to the two-ring product **7a**, which, upon deselenation, gives **8**, the major component of the reaction mixture. If diselenide **2a** is still present in the reaction mixture, it reacts with **8** to give the three-ring product **9**. These new paths make 2-seleno-uracil unsuitable as a model for the analysis of oxidative stress in transfer RNAs. On the other hand, only excess hydrogen peroxide causes oxidation of S2Ura. This process proceeds via sulfenic (Ura-SOH), sulfinic (Ura-SO_2_H), and sulfonic (Ura-SO_3_H) intermediates, and finally, after the elimination of sulfur oxides (SO_n_, *n* = 2 or 3), uracil is restored. Thus, we conclude that 2-selenouracil is oxidized more easily than 2-thiouracil, and the dominating oxidation path depends on (i) the concentration of reactants, (ii) excess oxidant, and (iii) the pH of the reaction mixture. Moreover, since the two- and three-ring products may be potential metabolites of degradation/oxidation of selenium-modified transfer RNAs, it is of interest to elaborate their influence on cell biology.

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
