# Peer review of "Different Oxidation Pathways of 2-Selenouracil and 2-Thiouracil, Natural Components of Transfer RNA"

_ijms, 2020, doi:10.3390/ijms21175956_

Round 1

Reviewer 1 Report

In this work the authors have investigated different oxidation pathways of 2-selenouracil and 2-thiouracil, which are natural components of transfer RNA, through reaction with hydrogen peroxide under different concentrations of reactant and oxidant and under different pH conditions.

The authors have performed a thorough analysis of reaction progress and characterisation of reaction products using NMR spectroscopy, ultra-performance liquid chromatography coupled with a high-resolution mass spectrometry and ultraviolet spectroscopy.

I suggest that the manuscript should be accepted for publication in International Journal of Molecular Sciences following some minor revisions to the text, as follows:

Page 1

Line 40.  Change ‘to suggest’ to ‘the suggestion’

Page 2

Line 50.  Change ‘undergoes’ to ‘proceeds’

Line 71.  Change ‘presented here work’ to ‘work presented here’

Line 74.  Remove the ‘a’ before ‘high’

Page 3

Line 113.  Insert ‘a’ before ‘selenium’

Page 7

Line 186.  Change ‘to identify’ to ‘identification of’

Page 12

Line 348.  Insert ‘a’ before ‘reducing’

Page 14

Line 412.  Change ‘mention’ to ‘mentioning’

Page 15

Line 446.  Remove ‘a’ before ‘700’

Line 453.  Change ‘by’ to ‘using an’

Line 455.  Insert ‘a’ before ‘SYNAPT’

Page 16

Line 479.  Insert ‘a’ before ‘Specord’

Line 501.  Remove ‘A’

Author Response

Dear Reviewer,

All your corrections have been introducted into the text as suggested. Thank you very much for improving the wording of our manuscript. We would like to mention that the text was language-edited by professional American Journals Experts office.

Reviewer 2 Report

The present manuscript describes the behaviour of 2-selenouracil under conditions of oxidation in comparison with its 2-thio counterpart. The authors identified intermediates and final products at several time points using NMR and UPLC-PDA-ESI-HRMS measurements and proposed putative transformation pathways to characterize the redox properties of selenium- and sulfur-containing uracils. This is a very thorough study demonstrating that the dominant oxidation path depends on the concentration of reactants, excess oxidant, and the pH of the reaction mixture. I only miss some 77Se NMR studies of the ensuing compounds but the evidence presented even in this form is satisfactory. The paper can be published after correcting some minor issues listed hereunder.

Minor issues:

Main text:

Lines 319-320: "2-Selenouracil (1a, Se2Ura) and 2-thiouracil (1b, S2Ura) are relatively rare nucleobases present in transfer RNAs" - there should be a reference to this fact, e.g. the RNA modification database (https://mods.rna.albany.edu/mods/) or other source.

Line 354: the reference to compd 7a is better done mentioning Scheme 1 instead of Figure 3.

Line 357: "Scheme 8A" should be "Scheme 8a".

Line 360: "transformed to described here two- and three-ring products" should be "transformed to two- and three-ring products described herein".

Lines 395-400: It is claimed that the carboxy group has a strong EWG effect. I would remove the epithet "strong": the pK difference (0.07) or K difference (17%) does not justify this.

Line 420: "Of note" should be replaced by "It is noteworthy that".

Line 469: "90°C" should be replaced by "90 °C".

P. 17, line 535: Ref. 1: the underlining for the editor's name is not justified. In general, the reference style should be checked (e.g. the author names should follow the convention "Hondal, R.J." and not "Hondal R.J."), cf. https://www.mdpi.com/journal/ijms/instructions.

Supporting info:

P. 2, Synthesis of 2-selenouracil: sodium is not soluble in toluene, at 120 °C it will only melt and it can be dispersed. The name of the intermediate (ethylformyl acetate) is not correct, it should be replaced by "ethyl 3-oxopropanoate".

The 700 MHz frequency for the NMR instrument would be valid only for the 1H nucleus, for the 13C nucleus 176 MHz should be listed.

It is embarassing that the average molecular weight is given for the compounds while the HRMS spectra show obviousy exact masses.

Author Response

Dear Reviewer,

All your corrections have been introduced into the text as suggested. Thank you very much for improving the wording of our manuscript and for indicating some of our errors. We introduced two references - 20 and 21 - related to the database of modified nucleosides, and therefore the referenced 20-31 had to be shifted to ref. 22-33. We would like to mention that the text was language-edited by professional American Journals Experts office.

The changes and corrections are underlined in red.